:ᐧᐧᐧPLOS ONE

# Complete chloroplast genomes of two *Siraitia* Merrill species: Comparative analysis, positive selection and novel molecular marker development

**Hongwu Shi**[1☯], **Meng Yang**[1☯], **Changming Mo**[2], **Wenjuan Xie**[3], **Chang Liu**[1], **Bin Wu**[1]\*, **Xiaojun Ma**[1]\*

**1** Institute of Medicinal Plant Development, Chinese Academy of Medical Sciences and Peking Union Medical College, Beijing, China, **2** Guangxi Crop Genetic Improvement and Biotechnology Laboratory, Nanning, China, **3** Guilin Medical University, Guilin, China

☯ These authors contributed equally to this work.
\* bwu@implad.ac.cn (BW); mayixuan10@163.com (XM)

**Data Availability Statement:** The chloroplast genome sequencing data were deposited in GenBank under accession numbers MK755853 and MK755854, respectively.

## Abstract

*Siraitia grosvenorii* fruit, known as Luo-Han-Guo, has been used as a traditional Chinese medicine for many years, and mogrosides are its primary active ingredients. Unfortunately, *Siraitia siamensis*, its wild relative, might be misused due to its indistinguishable appearance, not only threatening the reliability of the medication but also partly exacerbating wild resource scarcity. Therefore, high-resolution genetic markers must be developed to discriminate between these species. Here, the complete chloroplast genomes of *S. grosvenorii* and *S. siamensis* were assembled and analyzed for the first time; they were 158,757 and 159,190 bp in length, respectively, and possessed conserved quadripartite circular structures. Both contained 134 annotated genes, including 8 rRNA, 37 tRNA and 89 protein-coding genes. Twenty divergences (Pi > 0.03) were found in the intergenic regions. Nine protein-coding genes, *accD*, *atpA*, *atpE*, *atpF*, *clpP*, *ndhF*, *psbH*, *rbcL*, and *rpoC2*, underwent selection within Cucurbitaceae. Phylogenetic relationship analysis indicated that these two species originated from the same ancestor. Finally, four pairs of molecular markers were developed to distinguish the two species. The results of this study will be beneficial for taxonomic research, identification and conservation of *Siraitia* Merrill wild resources in the future.

## Introduction

*Siraitia* plants are important perennial vines belonging to the fourth most economically important plant family, Cucurbitaceae, and the genus has been widely cultivated as economic crops in southern China and northern Thailand [1]. Among these crops, *S. grosvenorii*, a traditional Chinese medicinal plant native to Guangxi, China, has been cultivated for approximately 200 years. The fruit of *S. grosvenorii*, Luo-Han-Guo, has been used as a traditional

**Funding:** This work was supported by Beijing Municipal Natural Science Foundation (5172028), the National Natural Science Foundation of China (81373914, 81573521), and CAMS Innovation Fund for Medical Sciences (CIFMS) (No. 2017-I2M-1-013).

**Competing interests:** The authors have declared that no competing interests exist.

Chinese medicine for the treatment of lung congestion, cold, and sore throat [2,3]. The primary active ingredients of *S. grosvenorii* are mogrosides, which are a class of cucurbitane-type triterpenoids, including mogrosides IV, V and VI. Modern pharmacology studies have shown that mogrosides have antidiabetic, antioxidative and anti-inflammatory effects [4,5]. Mogrosides are also natural zero-calorie sweeteners and have been used as sugar substitutes; mogroside V is 250 times sweeter than sucrose [6]. Compared to *S. grosvenorii*, *S. siamensis* has advantages in disease resistance and fruit set percentage [7]. Siamenoside I, a kind of mogroside, was separated from the fruit of *S. siamensis* and is approximately 560 times sweeter than sucrose [8], and is about 1.4 fold sweeter than aspartame. In recent years, people have paid more attention to developing and utilizing *Siraitia* germplasm resources because of the importance of mogrosides in sweetener development.

Many studies have focused on the improvement of cultivated varieties and excavation of the potential medicinal value of medicinal plants [9–11], as well as identification of the wild species [12,13]. Among *Siraitia* plant materials, only *S. grosvenorii* fruit has been stipulated to have medicinal use and is listed in the latest edition of the Chinese Pharmacopoeia [14]. Thus, indiscriminate use of wild relatives, such as *S. siamensis*, might cause Luo-Han-Guo's poor therapeutic effect. On the other hand, both *S. grosvenorii* and *S. siamensis* are dioecious and have a low natural pollination rate, leading to few fruits, although seed traits are important indicators for identifying these two species [15]. Most of the *Siraitia* plants origin privately from wild resources without professional identification and named with ordinary variety names [16]. These phenomena suggest that the cultivation of *Siraitia* species has been immethodical and nonstandard on some level. Moreover, the lack of an effective approach for distinguishing among *Siraitia* species has hindered genetic diversity studies and at least partly led to the gradual loss of some varieties. Thus, high-resolution molecular markers are urgently needed to solve these problems.

Universal molecular markers, such as *ITS*, *rbcL* and *psbA*, are widely used for identifying some species rapidly and accurately [17–19], but they cannot distinguish wild relatives. The chloroplast is a vital and semiautonomous plant cell organelle and has essential roles in photosynthesis and carbon fixation [20,21]. Although most plant chloroplast genomes display highly conserved structures, some structural rearrangements, including inverted repeat (IR) loss, gene loss and indels, are the result of adaptation to their environments [22]; thus, several highly variable regions could be developed as markers for species identification [23], such as indel and single nucleotide polymorphism (SNP) markers for *Panax ginseng* subspecies [24] and indel markers for *Ipomoea nil* and *Ipomoea purpurea* [25]. In addition, abundant closely related species could be identified by combining several markers, such as two indel markers for the identification of three *Aconitum* species [26].

In this study, the complete chloroplast genomes of *S. grosvenorii* and *S. siamensis* were assembled and analyzed, providing the first two sequences in *Siraitia* species. Comparative analyses revealed that the IR regions and coding sequence regions are highly conserved, and several higher-variation regions were primarily located in intergenic regions. Phylogenetic relationship analysis supported the position of two species in the basal lineage of Cucurbitaceae. The identification of nine protein-coding genes in several sites undergoing positive selection contributes to further investigation on the adaptive evolution of plants in ecosystems. Finally, four novel molecular markers (GSPC-F/R, GSPR-F/R, GSPB-F/R and GSPY-F/R) were developed to distinguish the two species. Overall, the sequencing and analysis of two species of chloroplast genomes in *Siraitia* will be beneficial for enhancing medicinal safety and for the species identification and conservation of wild *Siraitia* species, and it will provide new insight for the understanding of plant adaptive evolution in ecosystems.

## Materials and methods

### Plant materials, DNA extraction, and sequencing

The fresh leaves of two-year-old *S. grosvenorii* and *S. siamensis* plants were collected from Guangxi Medicinal Botanical Garden of the Institute of Medicinal Plant Development, Chinese Academy of Medical Sciences and Peking Union Medical College (Nanning, China), then frozen at -80˚C until further use. Total DNA was extracted from approximately 100 mg samples using a plant genomic DNA kit (DP305) (Tiangen Biotech Co., Ltd., Beijing, China), DNA quality was assessed in a Nanodrop 2000 spectrophotometer (Thermo Scientific), and DNA integrity was evaluated using a 1.0% (w/v) agarose gel. DNA samples from each species were used to prepare two separate libraries with an average insert size of 500 bp and sequenced using an Illumina HiSeq 4000 (Illumina Inc., San Diego, CA, USA) with a standard protocol.

### Chloroplast genome assembly

First, the low-quality reads sequenced from all the samples were filtered by Trimmomatic software [27]. Then, the trimmed reads, including nuclear and organelle genome data, were used to assemble the chloroplast genome. All chloroplast genomes of plants assessed in the National Center for Biotechnology Information (NCBI) were used to search against Illumina paired-end reads using SRA-BLASTN with an E-value cutoff of 1e-5 [28]. Clean reads with high homology were considered plastome reads and used for downstream genome assembly. SPAdes (v3.10.1) and CLC Genomics Workbench (v7) were used for the de novo genome assembly, the SPAdes using for the assembling that the parameters were set as "-k 21,33,55,77,99,127 –careful" [29]. The contigs obtained were identified by Gepard and spanned the entire plastome [30]. All the identified contigs were assembled using the SeqMan module of DNASTAR (v11.0) [31]. Then, three scaffolds, including the large single-copy (LSC), the IR, and small single-copy (SSC) regions, were obtained. The specific de-novo genome assembler NOVOPlasty was also used to reassemble the two species chloroplast genome for verification [32].To verify the assembly accuracy, the four boundaries between the single-copy (SC) regions and IR regions of the assembled sequence were confirmed by PCR amplification and Sanger sequencing, and the sequences of the primers are listed in S1 Table.

### Genome annotation, repeats and simple sequence repeats (SSRs) analyses

The online program Dual Organellar GenoMe Annotator (DOGMA, http://dogma.ccbb. utexas.edu/) and the Chloroplast Genome Annotation, Visualization, Analysis, and GenBank Submission (CPGAVAS) were used to annotate the two genomes [33,34]. The protein-coding sequences were verified by Blastp against the GenBank database. The tRNA genes were identified by tRNAscan-SE and DOGMA [33,35]. Then, manual corrections of the positions of the start and stop codons and the intron/exon boundaries were performed based on the entries in the plastome database using the Apollo program (v.1.11.8) [36,37]. The circular genomic maps were drawn using OrganellarGenomeDRAW (v1.2) with the default setting and checked manually [38]. The newly generated complete chloroplast genome sequences were submitted to GenBank.

The software CodonW (1.4.4) was used to investigate the distribution of codon usage with the relative synonymous codon usage (RSCU) ratio [39]. The codon usage frequency and GC content of both species were calculated using the programs Cusp and Compseq in EMBOSS (v.6.3.1) [40,41]. Repeats, including forward, palindromic, reverse, and complement, were identified by REPuter with the following setting parameters: 3 for Hamming distance and 30 for minimal repeat size [42]. SSRs were detected by MISA software with the parameters set as

reported previously, and the cutoffs of the unit numbers for mono-, di-, tri-, tetra-, penta-, and hexa- nucleotides were 8, 4, 4, 3, 3, and 3, respectively [43].

## Comparative genomic and selective pressure analyses

The mVISTA program in Shuffle-LAGAN mode with default parameters was used to compare the five complete chloroplast genomes using *S. grosvenorii* chloroplast genomes as a reference [44,45]. The sequence divergence of the chloroplast genomes was analyzed with a sliding window using DnaSP (v5.10) [46], and the step size was set to 200 bp with a 600 bp window length. Moreover, a total of 104 intergenic regions and 77 exons were manually extracted among four species, and the corresponding sequences aligned using ClustalW2 (v2.0.12) [47] were used to calculate the nucleotide variability (Pi) using DnaSP (v5.10). Selective pressure was analyzed for consensus protein-coding genes among twelve genomes from Cucurbitaceae species. Easy-CodeML software with the site model was performed to calculate the nonsynonymous (*Ka*) and synonymous (*Ks*) substitution ratios and likelihood ratio tests (LRTs). The values of both *Ka*/*Ks* (ω) and the LRTs were coupled to evaluate the selection on amino acid sites [48].

## Phylogenetic analyses

A total of 30 chloroplast genomes, including 28 from Cucurbitaceae, were used for the phylogenetic analyses, including *Nicotiana tabacum* and *Arabidopsis thaliana* as outgroups. In this study, 28 chloroplast genome sequences were downloaded from NCBI GenBank (S2 Table). The software MAFFT was used to generate alignments of 64 consensus protein-coding gene sequences [49], and then the alignments were manually adjusted by BioEdit [50]. Maximum likelihood (ML) analysis was carried out based on the Tamura-Nei model using a heuristic search for initial trees that were the most appropriate by Modeltest 3.7 [51]. Maximum parsimony (MP) analysis was conducted using PAUP (v4.0a) [52]. Bootstrap analysis was performed with 1000 replicates. Finally, the reconstructed trees were visualized using Figtree (v1.4.3) (http://tree.bio.ed.ac.uk/software/figtree/).

## Molecular marker development and validation

The molecular regions between the two *Siraitia* species were examined by the alignment and comparison of mVISTA similarities. The primers for the molecular markers were designed using the software Primer Premier 5.0 [53]. The accuracy of the molecular markers was verified using PCR amplification, and PCR was conducted with the following program: initial denaturation at 94˚C for 2 minutes; followed by 35 cycles of amplification at 94˚C for 20 seconds, 56˚C for 20 seconds, and 72˚C for 2 minutes; and final extension at 72˚C for 10 minutes. The PCR products were separated with 1.0% (w/v) agarose gel for 20 minutes at 180 volts. Then, the DNA fragments were purified and sequenced.

## Results and discussion

### General features of the chloroplast genomes

The chloroplast genome structures of the two species are similar to those of other Cucurbitaceae species, which display a single circular molecule with a typical quadripartite structure. The complete chloroplast genome of *S. grosvenorii* was 158,757 bp in length and consists of a pair of IRs (IRa and IRb, each 26,288 bp in length) separated by a LSC (87,625 bp) region and a SSC (18,556 bp) region (Fig 1, S1 Fig and S3 Table). The complete chloroplast genome of *S. siamensis* possessed the same structure. The IRa and IRb were 26,289 bp each, the LSC was 88,069 bp, and the SSC was 18,543 bp. In total, the length of the whole genome was 159,190

bp. (Fig 1, S3 Table). The length of each region is similar to those of most plant chloroplast genomes reported previously [54]. The sequencing data of *S. grosvenorii* and *S. siamensis* were deposited in GenBank under the accession numbers MK755853 and MK755854, respectively.

Further analysis results revealed that the two species had approximately 36.8% GC content (S3 Table), which was distributed unevenly across the whole chloroplast genome. In contrast to the LSC regions, the GC contents in the IR regions displayed a higher value across the whole chloroplast genome, 42.8% in both *S. grosvenorii* and *S. siamensis*, possibly resulting from rRNA genes (*rrn16*, *rrn23*, *rrn4.5* and *rrn5*, 55.3% GC content in both cases) with high GC content located in the IR regions [55]; the higher GC content in the IR region was regarded as an indicator of species affinity [56]. Moreover, the LSC regions had a GC content of 34.6% in both species, and the lowest values of 30.5% and 30.7% were found in the SSC regions in *S. grosvenorii* and *S. siamensis*, respectively. In addition, the GC contents of the protein-coding regions were 37.8% in *S. grosvenorii* and 37.9% in *S. siamensis*, and the percentages of GC content for the first, second, and third codon positions were 45.6%, 38.0% and 29.8% in

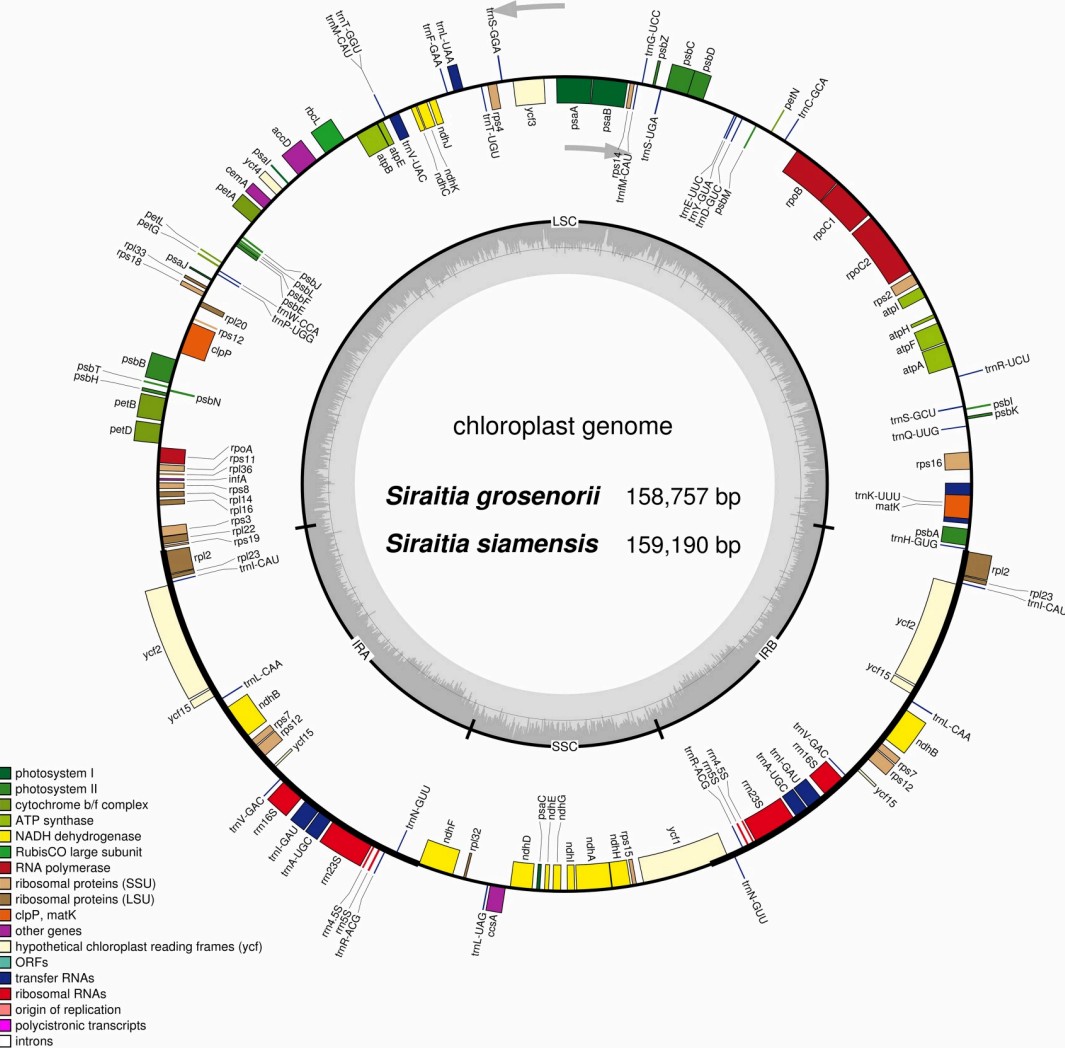

**Fig 1. Circular Gene map of the complete chloroplast genomes of *S. grosvenorii* and *S. siamensis*.** The quadripartite structure includes two copies of an IR region (IRa and IRb) that separated by (LSC) and SSC regions. Genes drawn in the circle are the transcribed clockwise, and those on the outside are transcribed counter-clockwise. The darker gray area in the inner circle show the GC content, whereas the lighter corresponds to AT content. Different genes groups are colored.

*S. grosvenorii* and 45.6%, 62.0% and 29.9% in *S. siamensis*, respectively (S3 Table). A bias toward using thymine (T) and adenine (A) in the third codon position has also been observed in other land plant plastomes [57–59]. The skewed GC distribution across the whole genome might be associated with the position of the origin and terminals for gene replication [60–62].

The two species exhibited the same gene content and arrangement in the chloroplast genome. A total of 134 genes were identified, including 89 protein-coding genes, 37 transfer RNA (tRNA) genes and four ribosomal RNA (rRNA) genes. Eight protein-coding genes, seven tRNA genes, and four rRNA genes were duplicated in the IR regions in both species (Fig 1); *ycf15-orf* was duplicated, and its start codon was GTG. The data revealed that 23 genes contain introns in both genomes, 21 with one intron and the rest with two introns, and the genes *clpP* and *ycf3* both contained three exons (S4 and S5 Tables). The gene *clpP* was related to energy transformation [63], and *ycf3* was necessary for the stable accumulation of photosystem I complexes [64]. Introns found in functional genes play a significant role in the regulation of gene expression, which can trigger desirable biological traits at particular times [65,66]. In addition, the *rps12* gene contained three exons and one intron because of trans-splicing, which resulted in a 5' end exon located in the LSC region, whereas the remaining exons were located in the IRs (S5 Table). Therefore, *rps12* was duplicated in the IR region. Furthermore, the results of gene location analysis revealed twelve genes with partial overlaps in their sequences: *trnK-UUU/ matK*, *psbD/psbC*, *trnM-CAU/trnT-GGU*, *atpE/atpB*, *rps3/rpl22*, and *trnP-UGG/trnP-GGG*.

The basic characteristics of the chloroplast genomes of two *Siraitia* species and six other species from Cucurbitaceae are shown in S6 Table. Comparative analysis showed that the lengths of the eight genomes ranged from 155,293 bp (*Cucumis sativus*) to 159,190 bp (*S. siamensis*), and the overall GC content percentage of *C. sativus* (37.2%) was higher than that of any other genome (36.7%-37.1%). However, little difference could be found in gene number, gene type or GC content between *S. grosvenorii* and *S. siamensis*, which suggests that we should focus on other areas to find variation, such as intergenic spacers.

## IR/SC boundaries and IR contraction and expansion

The contraction and expansion of the IR regions account for common evolutionary events and are a major cause of differences in chloroplast genome size, and evaluating them could shed some light on the evolution of some taxa [67,68]. A detailed comparison of the IR/SC boundary regions of twelve species is shown in Fig 2. *A. thaliana* and *N. tabacum* were set as outgroups, and the rest were Cucurbitaceae. From the comparison, we noticed that the lengths of *ycf1* pseudogenes of both *Siraitia* species were both 1,181 bp, almost as long as those of the other five Cucurbitaceae species, except *Lagenaria siceraria* (28 bp) and *Momordica charantia* (29 bp). In addition, the *ndhF* gene, located in the SSC, reaches 134–135 bp across the IRb/SSC boundary in both *L. siceraria* and *M. charantia*, while the corresponding region was 7–12 bp in most other Cucurbitaceae species. Moreover, the gene *rps19* is located in the LSC region with 265–277 bp across the LSC/IRb boundary in *Trichosanthes kirilowii*, *Hemsleya lijiangensis* and *Gynostemma laxiflorum*, whereas in other species, it was located away from the LSC/IRb by 68–208 bp. The *rpl2* gene, duplicated in the IRs in most species, was present as only one copy in the IRb region in *M. charantia*, *Cucurbita pepo* and *Citrullus lanatus*, which might result from the location of the LSC/IRb boundary. All of these phenomena were related to the contraction/expansion of two IR regions in the complete chloroplast genomes.

## Codon usage

RSCU is a measure of nonuniform synonymous codon usage in coding sequences in which values above 1 indicate that codons are used more frequently than expected [69,70]. All the

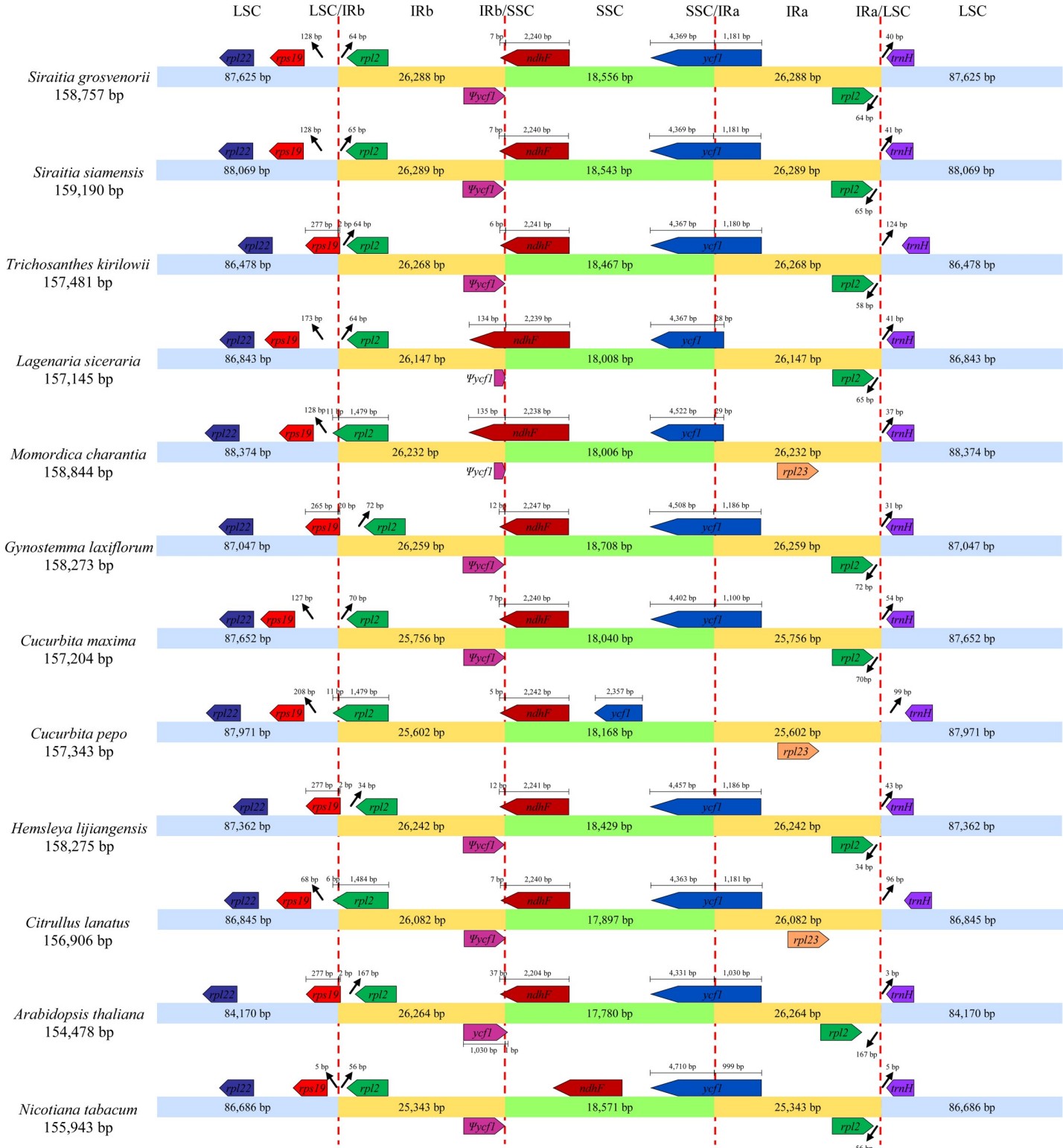

**Fig 2. Comparison of border distance between adjacent genes and junctions of the LSC, SSC and IRs regions among twelve chloroplast genomes.** Numbers with arrow above the gene features mean the distance between the ends of genes and the borders sites. The figure is not to scale with respect to sequence length.

protein-coding genes were encoded with 26,509 and 26,508 codons in the *S. grosvenorii* and *S. siamensis* chloroplast genomes, respectively. Detailed codon analysis revealed that the two cases had similar codon constitutions and RSCU values (S7 Table). Among these codons, leucine (Leu) and cysteine (Cys) were, respectively, the highest (10.48%) and lowest (1.13%) prevalence amino acid codons in the two *Siraitia* species. In addition, most of the amino acid codons showed preferences, with exception of methionine (Met) and tryptophan (Trp), which both had RSCU values of 1. The chloroplast genomes of the two *Siraitia* species both had 30 biased codons with RSCU >1, and the third positions of the biased codons were A/U except for Leu (UUG); otherwise, the codons with high frequency (>30%) and fraction were Asp-GAU, Glu-GAA, Ile-AUU, Lys-AAA, Leu-UUA, Asn-AAU, and Tyr-UAU, and the bias toward these seven codons was consistent with the low content of GC in the third codon position. Fig 3 shows that the RSCU value increased with the quantity of codons coding for a specific amino acid. A strong AT bias in codon usage is common in sequences with strong codon preference and was also found in most other land plant chloroplast genomes [71,72].

## Repeat structure and SSRs analyses

Repeat units, which are distributed in chloroplast genomes with high frequency, play an important role in genome evolution [73–75]. S2(A) Fig shows repeat structures that were longer than 30 bases in eight species. The repeats of the *S. grosvenorii* chloroplast genome consist of 19 forward, 24 palindromic, two reverse, and one complement. By contrast, a slightly different number of repeats was found in *S. siamensis*, which contained 17 forward, 20 palindromic, one reverse and no complement. In the eight-species comparison, the number of repeats for *C. lanatus* (31) was the lowest, and the highest number of repeats was 49 in *M. charantia*, *G. laxiflorum*, and *Cucurbita maxima*.

On the other hand, SSRs, which are also known as microsatellites and are distributed abundantly across genomes, are tracts of repetitive DNA with certain motifs, ranging from 1–6 or more base pairs, that are repeated typically 5–50 times [76,77]. SSRs are widely used as molecular markers for species identification, analysis of phylogenetic relationships and population genetics because of their high polymorphism rates and stable reproducibility [78,79]. Here, a total of 252 and 253 SSRs were identified by MISA software within the chloroplast genomes of *S. grosvenorii* and *S. siamensis*, respectively (S2(B) Fig), and mononucleotide repeats were largest in number, 57 and 56, respectively. Moreover, S8 Table shows that A/T mononucleotide repeats (97.2% and 97.8%, respectively) were the most common, and for dinucleotide repeats, AT/ TA (68.4% and 67.8%, respectively) was the majority. However, only one repeat unit (AAT/TTA) was found among trinucleotide repeats. In addition, AT-rich repetitive motifs were high in the remaining SSR types. The SSRs within the chloroplast genomes of both species mainly comprised AT-rich repetitive motifs, consistent with the fact that AT content (63.2%) was very high (GC content was 36.8%) in both cases. Furthermore, these results were also consistent with previous reports that proportions of short poly-A or poly-T repeats were higher than those of poly-G or poly-C within most SSRs in many plant chloroplast genomes [80,81]. Distribution of the SSRs loci in the chloroplast genome of *S. grosvenorii* and *S. siamensis* were exhibited in S9 Table.

## Sequence divergence and nucleotide diversity

Complete chloroplast genomes are often used to analyze plant taxonomy, phylogenetic relationships, and genetic diversity [82]. In this study, two *Siraitia* species were compared with other three species in Cucurbitaceae using mVISTA software, and *S. grosvenorii* was set as the

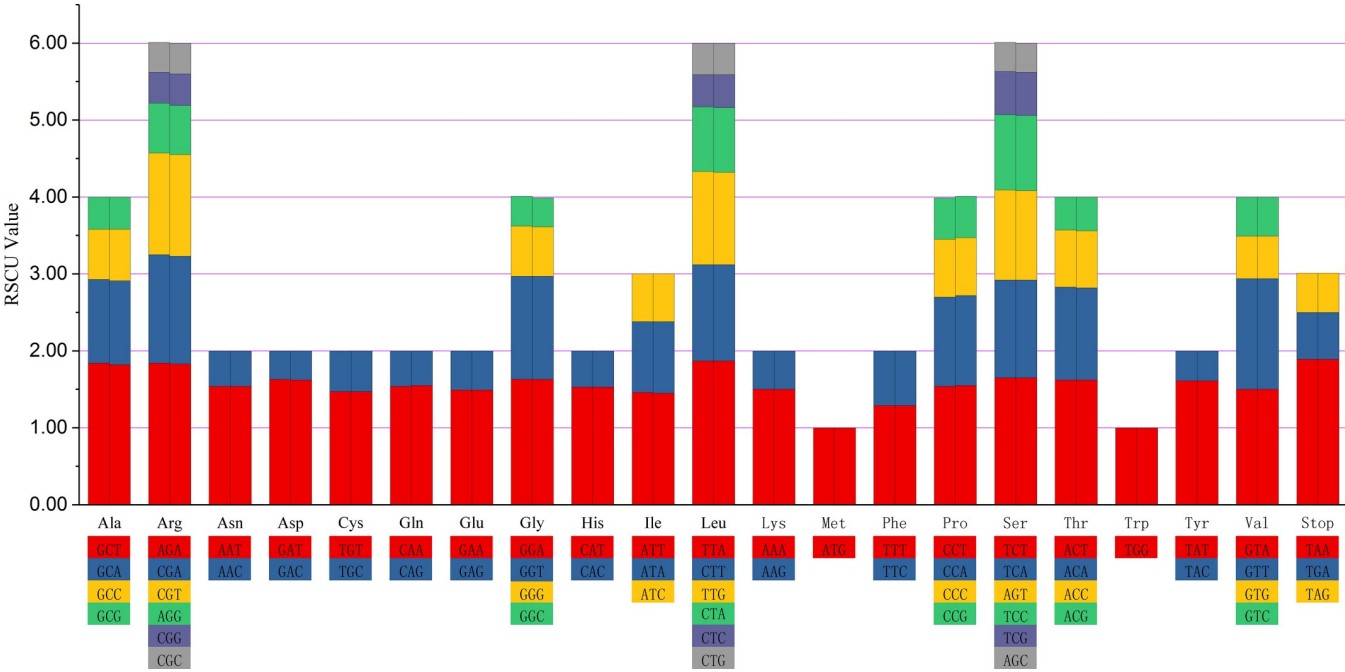

**Fig 3. Codon usage for 20 amino acid and stop codons in all protein-coding genes.** The columnar stacking diagram on the left and right of each amino acid display the codon usage within the chloroplast genome of *S. grosvenorii* and *S. siamensis*, respectively.

reference. As shown in Fig 4, sequence divergence was similar for the whole sequences of the complete chloroplast genomes. In contrast to the other two Cucurbitaceae species, *M. charantia* was more similar to the two *Siraitia* species across the complete chloroplast genomes. The data plot revealed that the noncoding region was more divergent than its coding counterparts. The two IR regions were both less divergent than the single-copy regions, which might be the result of the four highly conserved rRNAs located in the IR regions [55].

In addition, the nucleotide diversity of 181 regions was analyzed using DnaSP software, including 77 protein-coding genes and 104 intergenic regions among four chloroplast genomes (*T. kirilowii*, *M. charantia*, and two *Siraitia* species). The results revealed that intergenic regions were more divergent than protein-coding genes (Fig 5). The average nucleotide variability (Pi) in the intergenic regions was 0.01772, almost twice as much as that in protein-coding genes (Pi = 0.00950). This is consistent with previous research in angiosperm chloroplast genomes [83]. The *trnL-ccsA* (Pi = 0.07302) and *petG-trnW* (Pi = 0.06780) regions were notably variable among the intergenic regions, as were the genes *ycf1* (Pi = 0.03048), *psaJ* (Pi = 0.02972) and *atpE* (Pi = 0.02447) among the protein-coding genes. *ycf1* is commonly used as a representative plant DNA barcoding region [84]. Several highest-level divergences (Pi > 0.03) were found in the intergenic regions and could be developed as specific molecular markers for species identification, including *trnR-atpA*, *ndhC-trnV*, *petG-trnW*, *rpl32-trnL*, *trnL-ccsA*, *ndhF-rpl32*, *psbZ-trnG*, *psbC-trnS*, *ndhG-ndhI*, *rps8-rpl14*, *ccsA-ndhD*, *trnT-trnL*, *psbK-psbI*, *psbA-trnK*, *rps15-ycf1*, *trnF-ndhJ*, *atpA-atpF*, *rps19-rpl2*, *psaC-ndhE*, and *petD-rpoA*. It has been reported that divergent noncoding regions allow the discrimination of potential molecular markers and DNA barcodes [85]. Furthermore, sliding window analysis was also performed (S3 Fig). The average value of Pi was 0.00221 between the two *Siraitia* species, and higher variability was found in the LSC and SSC regions.

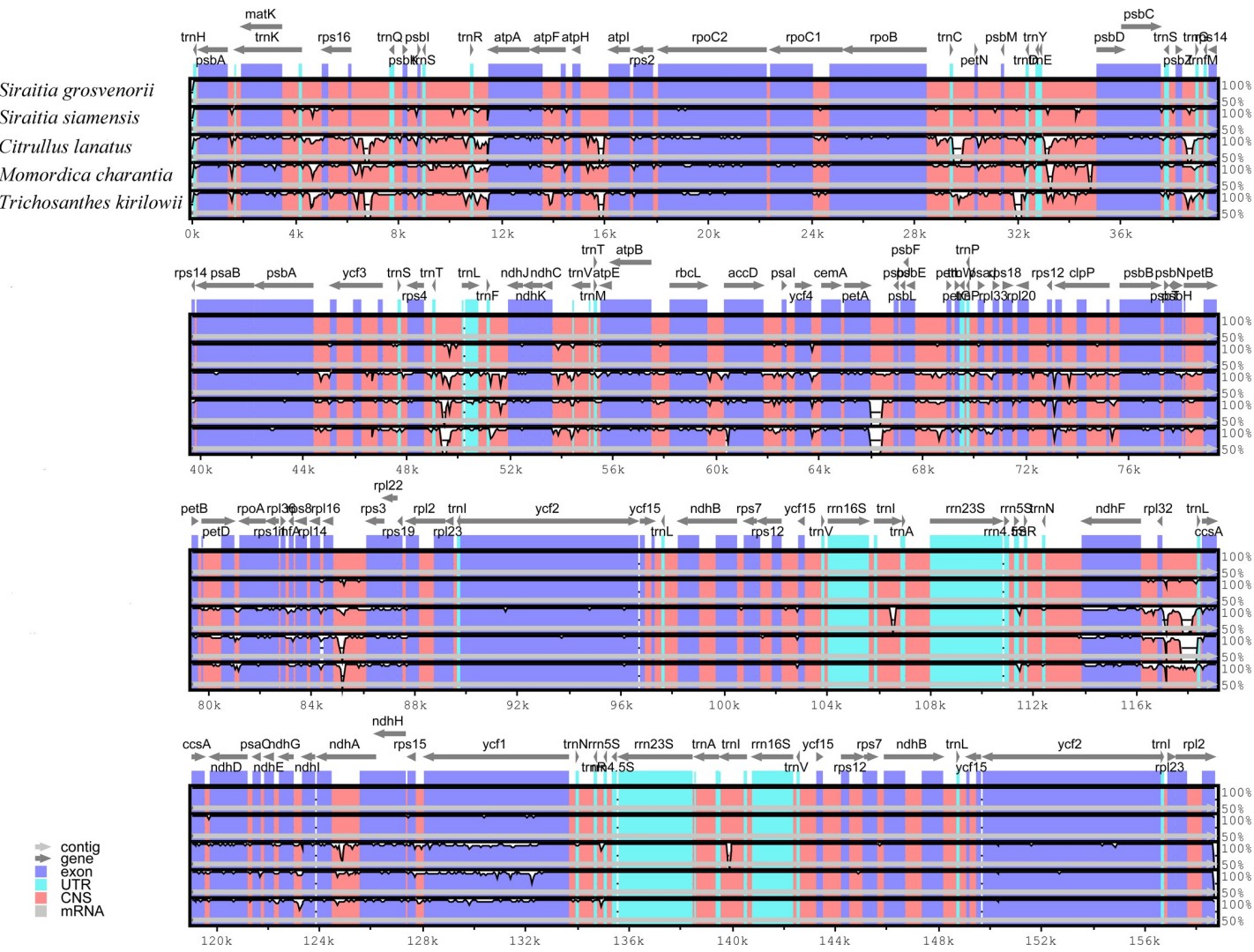

**Fig 4. Structure comparison of five chloroplast genomes using mVISTA program.** Gray arrows and thick black lines above the alignment indicate genes with their orientation and the position of the IRs, respectively. Purple bars, blue bars, pink bars, gray bars and white peaks represent exon, Untranslated Region (UTRs), Conserved Noncoding Sequences (CNS), mRNA and genomes differences, respectively. A cut-off of 70% identity was used for the plots, and the Y-scale represents the percent identity ranging from 50% to 100%.

## Phylogenetic relationships of six genera within Cucurbitaceae

Chloroplast genomes are significant genomic resources for the reconstruction of accurate and high-resolution phylogenetic relationships and taxonomic status in angiosperms [86]. Complete chloroplast genomes and protein-coding genes have been widely employed to determine phylogenetic relationships at almost every taxonomic level [87]. In this study, to identify the phylogenetic positions of the two *Siraitia* species within the Cucurbitaceae family, we aligned 64 protein-coding sequences from 30 chloroplast genomes; *A. thaliana* and *N. tabacum* were set as outgroups, and the alignment length was 62,522 bp. The ML and MP trees displayed similar phylogenetic topologies (Fig 6). All nodes in the ML tree and MP tree were strongly supported by high bootstrap values: 22 of 27 nodes with 100% bootstrap values were found in the MP tree and 21 of 27 in the ML tree. In addition, the results illustrated that two *Siraitia* species were the most closely related species to *M. kirilowii*, and these two taxa were grouped with two

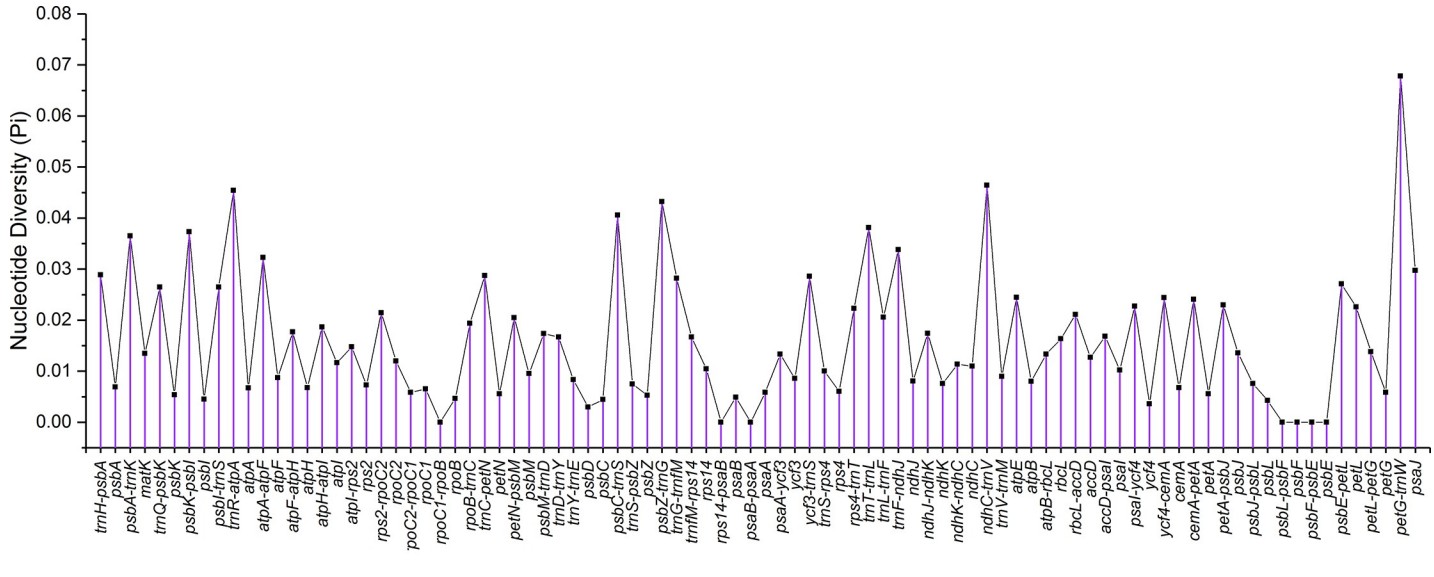

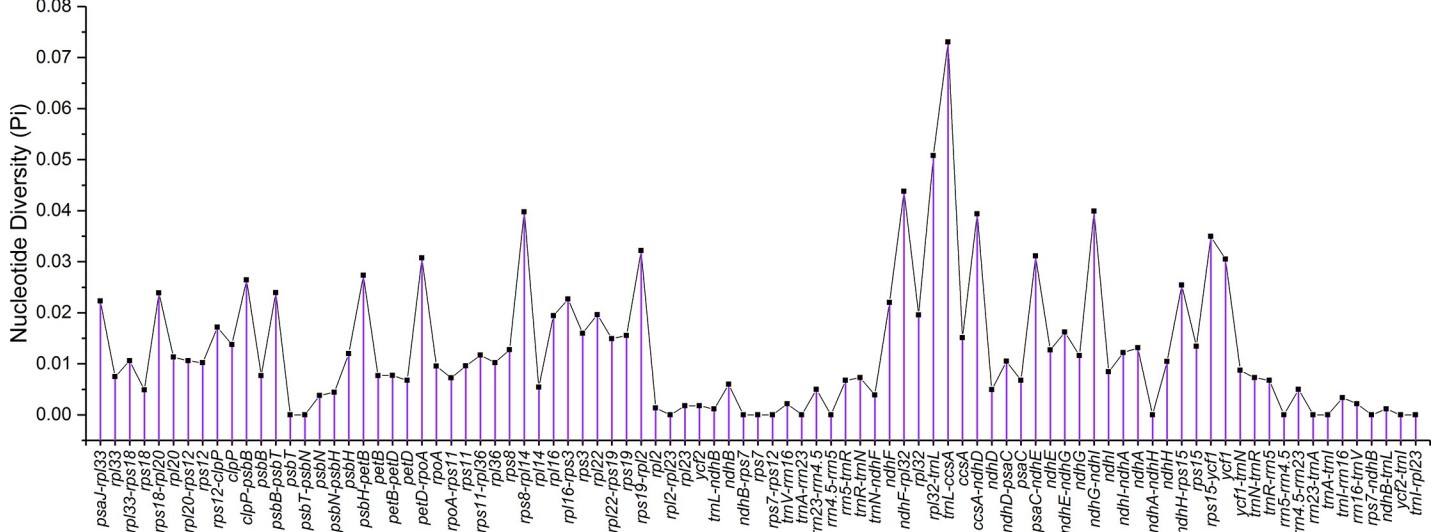

**Fig 5. Comparison of nucleotide diversity (Pi) value for 77 protein-coding genes and 104 intergenic regions among six closely species.**

species from Sicyoeae, three species from Cucurbiteae, and nine from Benincaseae, which showed a nested evolutionary relationship in the MP and ML trees. Furthermore, all species were clustered into a lineage distinct from the outgroups and strongly supported the new classification system of Cucurbitaceae [54].

## Selective pressure events

Synonymous substitutions (*Ks*) accumulate nearly neutrally, whereas nonsynonymous substitutions (*Ka*) are subjected to selective pressures of varying degree and direction (positive or negative); values of *Ka/Ks* (ω) above 1.0 indicate that the corresponding genes experience positive selection, while ω values ranging from 0.5 to 1.0 indicate relaxed selection [88]. In the current study, we performed a selective analysis of the exons of each protein-coding gene using site-specific models with four comparison models (M0 vs. M3, M1a vs. M2a, M7 vs. M8 and

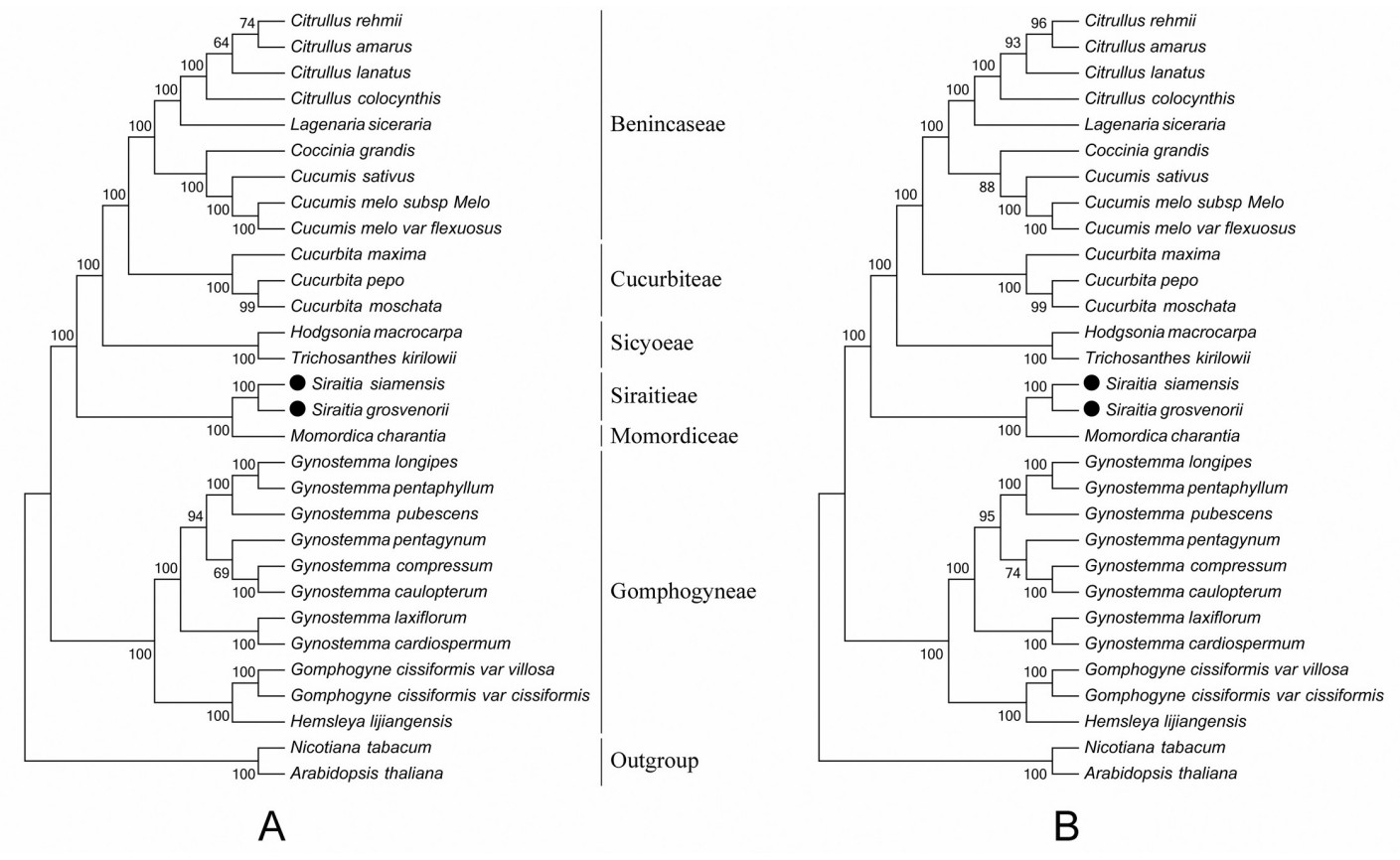

**Fig 6. Molecular phylogenetic relationship of Siraitieae with 64 protein-coding genes of 26 cucurbitaceae species.** The unrooted trees were constructed by (A) maximum parsimony (MP) and (B) maximum likelihood (ML) methods with bootstrap values ≥50%.

M7a vs. M8a, LRT threshold p ≤ 0.05) in EasyCodeML software as reported previously [48]. Among the eight models, M2a was the positive model, and p ($p_0$, $p_1$, $p_2$) were the proportions of purifying selection, neutral selection, and positive selection. A total of 58 consensus protein-coding genes from 12 closely related species were evaluated with respect to selective pressure. Nine genes (*accD*, *atpA*, *atpE*, *atpF*, *clpP*, *ndhF*, *psbH*, *rbcL, and rpoC2*) were found to have undergone positive selection, and the $\omega_2$ values ranged from 2.17 to 11.44 in the M2a model (Table 1).

To determine which sites were subject to positive selection, naive empirical Bayes (NEB) and Bayes empirical Bayes (BEB) methods were used to analyze the location of consistent selective sites in the alignment of chloroplast genomes in the M7 vs. M8 model. The data analysis revealed that the gene *rpoC2* possesses 7 positive selective sites, followed by *atpF* (6), *rbcL* (4), *atpA* (2), *atpE* (2), *clpP* (2), *ndhF* (2), *accD* (1), and *psbH* (1). All positively selected sites in these nine genes are shown in Table 1.

Among the nine positively selected genes, *rpoC2* encodes the RNA polymerase β". A comparison of *rpoC2* between fertile lines of sorghum and cytoplasmic male sterile lines showed that a 165 bp deletion was identified that encodes several protein motifs involved with transcription factors; this region might play an important role in the regulation of developmental pollination [89]. The *Siraitia* species are dioecious, so the finding that *rpoC2* evolved under positive selection might indicate that it is essential for sex differentiation. The chloroplast plays

**Table 1. The results of positive selective pressure analysis in M7 vs.M8 model.**

| Gene name | Modle | np | LnL | ω2 (M2a) | LRTs (2ΔLnL) | LRT P-value | Positive sites |
|---|---|---|---|---|---|---|---|
| *rpoC2* | M8 | 26 | -8422.120232 | 11.44045 | 43.1617 | 0 | 486 V*, 527 P**, 754 S*, 965 S**, 1024 F*, 1331 P**, 1353 F* |
| | M7 | 24 | -8443.701061 | | | | |
| *atpF* | M8 | 26 | -1276.119906 | 2.17251 | 6.8495 | 0.032557126 | 2 E*, 8 K**, 9 K*, 14 F*, 61 E*, 97 L* |
| | M7 | 24 | -1279.544665 | | | | |
| *rbcL* | M8 | 26 | -2716.450055 | 3.38881 | 38.4242 | 0.000000005 | 251 M*, 255 I**, 470 E*, 472 M* |
| | M7 | 24 | -2735.662163 | | | | |
| *atpA* | M8 | 26 | -2937.475433 | 5.46438 | 14.4739 | 0.00071952 | 258 S*, 484 N* |
| | M7 | 24 | -2944.712359 | | | | |
| *atpE* | M8 | 26 | -839.182861 | 7.38230 | 7.7637 | 0.020612244 | 36 D*, 130 G* |
| | M7 | 24 | -843.064731 | | | | |
| *ndhF* | M8 | 26 | -5697.671745 | 9.36793 | 15.6948 | 0.000390771 | 509 I*, 686 I* |
| | M7 | 24 | -5705.519135 | | | | |
| *clpP* | M8 | 26 | -1298.084585 | 3.91456 | 9.4032 | 0.009080555 | 11 V*, 51 Y* |
| | M7 | 24 | -1302.786205 | | | | |
| *accD* | M8 | 26 | -3041.598734 | 2.91222 | 9.2524 | 0.009791966 | 223 I* |
| | M7 | 24 | -3046.224927 | | | | |
| psbH | M8 | 26 | -387.753418 | 11.27276 | 10.1816 | 0.006153015 | 72 S** |
| | M7 | 24 | -392.844231 | | | | |

* $P < 0.05$

** $P < 0.01$

np represents the degree of freedom.

important roles in photosynthesis and carbon fixation, and six genes (*atpF*, *rbcL*, *atpA*, *atpE*, *ndhF*, and *psbH*) with essential roles in photosynthesis were positively selected in this study. The *Siraitia* species are distributed in southeast Asia, so requirements for sufficient light might have exerted strong selective forces on the six genes during plant evolution. This phenomenon was also found in species within the *Urophysa* genus, which is distributed in southwest China [75]. The *clpP* gene, encoding the ATP-dependent clp protease, is likely involved in the transformation of chloroplast protein and might be essential for shoot development under clpP-mediated protein degradation [63,90,91]. The positive selection of the gene *clpP* in our study might be associated with the evolution of the vining character within Cucurbitaceae. As for the gene *accD*, it encodes the β–carboxyl transferase subunit of acetyl-CoA carboxylase [92]; it is a vital gene for leaf development and has effects on leaf longevity and seed yield [93]. Expression of the gene *accD* might indirectly affect the efficiency of photosynthesis. These nine genes have undergone positive selection, which might be the result of adaptation to their barren environment.

## Molecular markers for distinguishing *S. grosvenorii* and *S. siamensis*

In this study, several notably variable regions were found in the comparison of the two chloroplast genomes. To develop high-resolution molecular markers for the identification of these two species, the specific divergent regions, including *ndhC-trnV-UAC*, *trnR-UCU-atpA*, *rpoB-trnC-GCA* and the gene *ycf1*, were chosen as molecular marker regions, and specific primers were designed against the conserved regions (Table 2). The primers, named GSPC-F/R, GSPR-F/R and GSPB-F/R, were used for amplifying the three intergenic regions and produced

**Table 2. Primer identification for molecular markers.**

| Primer name | Primer sequence (5' to 3') | position |
|---|---|---|
| GSPC-F | GATGAACCAAATCAAGTGGC | ndhC-trnV-UAC |
| GSPC-R | CAGAAGCAGGACGATAGAGA | |
| GSPR-F | GGTTCAAATCCTATTGGACG | trnR-UCU-atpA |
| GSPR-R | GGCAAGAGGTCAACGATTAC | |
| GSPB-F | CTGTTTCCTACTCACACGAG | rpoB-trnC-GCA |
| GSPB-R | GGATTGGCTCTATCTCTTCG | |
| GSPY-F | GACGACTTGCTTTAGCGTTG | ycf1 |
| GSPY-R | GGACTAAACAGGAACAAGAG | |

different-length fragments in *S. grosvenorii* and *S. siamensis*, respectively. The gene *ycf1*, with high divergence, was also chosen for the development of molecular markers, and several SNP sites were found after amplification with GSPY-F/R. (Fig 7). The four molecular markers developed in this study will contribute to the identification of *Siraitia* species and facilitate efficient utilization and conservation of wild *Siraitia* resources.

The lack of an effective approach made the position of different species in the genus unreliable until the *Siraitia* Merrill was acknowledged in 1980 [15]. The phylogenetic analysis in this study showed that the two *Siraitia* species were the most closely related species to *M. kirilowii*, which explains the reason that these species were placed into *Momordica* L. with the name of *Momordica grosvenorii* Swingle initially [94]. To the best of our knowledge, only seven species within the genus *Siraitia* have been confirmed based on morphological characteristics [15], although some varieties may remain undiscovered. Unfortunately, the increasing demand for high production of these species has brought the wild resources to the verge of depletion. The comparative analysis of chloroplast genomes in this study revealed that several highly variable intergenic regions will contribute to the development of specific markers to support the conservation of wild *Siraitia* varieties. Among the seven species, *S. grosvenorii*, which is recorded as both a medicinal and an edible species [4], has been widely cultivated as an important commercial crop. *S. siamensis* is known to have better disease resistance and setting percentage and considerable siamenoside I content. Different germplasm resources should be distinguished accurately and used for different purposes. Not only do adulterants of raw medicinal materials threaten the safety and reliability of food and medicine, but they also exacerbate the scarcity of wild resources in similar species. Our development of novel molecular markers will be of great use in the species authentication and conservation of *Siraitia* wild resources in the future.

## Conclusions

In the current study, two complete chloroplast genomes of the *Siraitia* genus were assembled and analyzed for the first time. In a comparison with other species within Cucurbitaceae, several highly divergent noncoding regions were identified that would be beneficial for developing high-resolution molecular markers. Phylogenetic relationship analysis supported that *S. grosvenorii* and *S. siamensis* originated from the same ancestor, consistent with previous studies. Furthermore, 9 protein-coding genes were found to undergo selection, which might be the result of adaptation to the environment. Finally, molecular markers (GSPC-F/R, GSPR-F/R, GSPB-F/R and GSPY-F/R) were developed to distinguish the two species. The results in this study will be beneficial for taxonomic research, species identification, and conservation of the genetic diversity of *Siraitia* wild resources in the future.

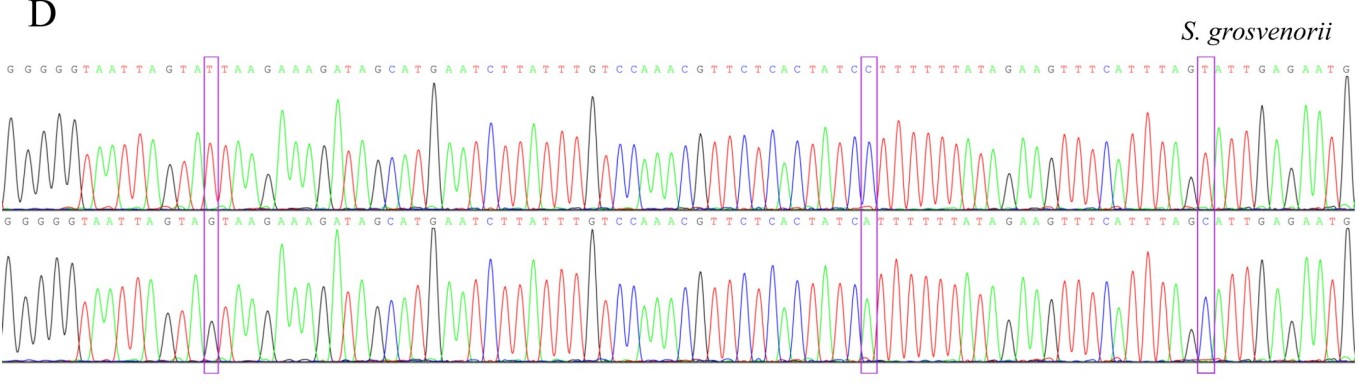

**Fig 7. Schematic diagram displayed the four novel molecular markers.** The indel makers in the intergenic spacers, including GSPC-F/R, GSPR-F/R, and SCPB-F/R, and ycf1 SNP were verified in *Siraitia* species with five individuals. (A) Sequencing results showed that PCR amplification by GSPC-F/R between the

*ndhC* and *trnV-UAC* in *S. grosvenorii* and *S. siamensis* were 790 bp and 808 bp, respectively. The difference can also be found by GSPR-F/R (B), and SCPB-F/R (C) between the two *Siraitia* species. (D) GSPY-F/R marker for *ycf1* SNP sites were validated in each case, which were effective to discriminate the two *Siraitia* species.

## Supporting information

**S1 Fig. Chloroplast genomes of S. grosvenorii and S. siamensis in linear form.**
(TIF)

**S2 Fig. Distribution of the number of different type repeats and SSRs.** (A) Repeat sequence in eight chloroplast genomes. Repeat sequences were identified by REPuter with length ≥30bp and sequence identified ≥90%. F, P, R, and C are the abbreviation of repeat type F (forward), P (palindrome), R (reverse) and C (complement), respectively. Different length repeat sequences are colored correspondingly. (B) Analysis of simple sequence repeat (SSRs) in chloroplast genomes of five species.
(TIF)

**S3 Fig. Sliding window analysis of the whole chloroplast genome.** Window length: 600 sites, Step size: 200 sites. X-axis: position of the midpoint of a window; Y-axis: nucleotide diversity (π) of each window. (A) Pi among *S. grosvenorii* and *S. siamensis*; (B) Pi among six species of Cucurbitaceae.
(TIF)

**S1 Table. Primer sequence at the boundaries between single cope and IR regions.**
(DOCX)

**S2 Table. List of chloroplast genome sequence used in the study.**
(DOCX)

**S3 Table. Base composition in the chloroplast genomes of S. grosvenorii and S. siamensis.**
(DOCX)

**S4 Table. Genes contained in the chloroplast genomes of S. grosvenorii and S. siamensis.**
(DOCX)

**S5 Table. Location information of genes with introns in the chloroplast genome of S. grosvenorii and S. siamensis.**
(DOCX)

**S6 Table. Comparisons among the chloroplast genome characteristics of S. grosvenorii and S. siamensis, and other six Cucurbitaceae species.**
(DOCX)

**S7 Table. Codon usage and codon-anticodon recognition in all protein-coding genes of the chloroplast genomes of two Siraitia species.**
(DOCX)

**S8 Table. Types and amounts of SSRs in the S. grosvenorii and S. siamensis chloroplast genomes.**
(DOCX)

**S9 Table. Distribution of the SSRs loci in the chloroplast genome of S. grosvenorii and S. siamensis.**
(XLSX)

## Acknowledgments

The authors acknowledge Mr. Jianguo Zhou, Miss. Hui Zhang and Miss. Mei Jiang in Institute of Medicinal Plant Development for their kind suggestions in the raw data process. We also thank the Guangxi Medicinal Botanical Garden of the Institute of Medicinal Plant Development, Chinese Academy of Medical Sciences and Peking Union Medical College (Nanning, China) for supplying the plant materials as a generous gift.

## Author Contributions

**Conceptualization:** Hongwu Shi, Chang Liu, Xiaojun Ma.

**Data curation:** Meng Yang, Bin Wu.

**Formal analysis:** Meng Yang, Bin Wu.

**Funding acquisition:** Xiaojun Ma.

**Investigation:** Changming Mo, Xiaojun Ma.

**Methodology:** Hongwu Shi, Meng Yang, Changming Mo, Wenjuan Xie, Chang Liu, Bin Wu.

**Project administration:** Changming Mo, Xiaojun Ma.

**Resources:** Changming Mo, Wenjuan Xie.

**Software:** Hongwu Shi, Meng Yang, Chang Liu, Bin Wu.

**Supervision:** Chang Liu, Bin Wu, Xiaojun Ma.

**Validation:** Chang Liu.

**Visualization:** Chang Liu, Xiaojun Ma.

**Writing – original draft:** Hongwu Shi, Meng Yang.

**Writing – review & editing:** Hongwu Shi, Bin Wu, Xiaojun Ma.

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
