## [Decision Letter · Decision Letter 0]

9 Sep 2019

PONE-D-19-14994

Complete chloroplast genomes of two Siraitia Merrill species: comparative analysis, positive selection and novel molecular marker development

PLOS ONE

Dear Dr Ma,

Thank you for submitting your manuscript to PLOS ONE. After careful consideration, we feel that it has merit but does not fully meet PLOS ONE’s publication criteria as it currently stands. Therefore, we invite you to submit a revised version of the manuscript that addresses the points raised during the review process.

Please make all the necessary changes, which is essential and improve your manuscript by incorporating all the comments suggested by three reviewer before submitting the revised version.

We would appreciate receiving your revised manuscript by November 7, 2019. To enhance the reproducibility of your results, we recommend that if applicable you deposit your laboratory protocols in protocols.io, where a protocol can be assigned its own identifier (DOI) such that it can be cited independently in the future. For instructions see: http://journals.plos.org/plosone/s/submission-guidelines#loc-laboratory-protocols

We look forward to receiving your revised manuscript.

Kind regards,

Shashi Kumar, Ph.D.

Academic Editor

PLOS ONE

Journal Requirements:

Additional Editor Comments (if provided):

Dear Dr. Ma,

Please correct your manuscript following the comments of three revivers.

Reviewers' comments:

Reviewer's Responses to Questions

**Comments to the Author**

1. Is the manuscript technically sound, and do the data support the conclusions?

Reviewer #1: Yes

Reviewer #2: Yes

Reviewer #3: Yes

2. Has the statistical analysis been performed appropriately and rigorously? 

Reviewer #1: Yes

Reviewer #2: Yes

Reviewer #3: Yes

3. Have the authors made all data underlying the findings in their manuscript fully available?

Reviewer #1: Yes

Reviewer #2: Yes

Reviewer #3: Yes

4. Is the manuscript presented in an intelligible fashion and written in standard English?

Reviewer #1: Yes

Reviewer #2: Yes

Reviewer #3: Yes

5. Review Comments to the Author

Reviewer #1: The paper assembles and analyses two complete chloroplast genomes of the Siraitia genus. Paper covers most of the aspects of comparative genome analysis between the two Siraitia genus and have developed molecular markers to distinguish the two species. Overall the paper is well written, however there are few suggestions mentioned below:

1) Line 46: There is no reference to the statement written. Please add reference to the this statement.

2) Line 56: "Unstable treatment effect" statement is not clear.

3) Line 62: "most origin plants" which origin plants author is mentioning, statement is not clear.

4) Line 168: Please mention the version of Modeltest used.

5) Line 197: Please add some description of the figure shown " The quadripartite structure includes two copies of an IR region (IRA and IRB) that separate large single-copy (LSC) and small single copy (SSC) regions ".

6) Line 199: It would be easier if you could represent the chloroplast genome in linear form also for better and faster visualization.

7) Line 230: Add reference of "S5 Table" next to line " In addition, the rps12 gene contained three exons and one intron because of trans-splicing, which resulted in a 5’ end exon located in the LSC region, whereas the remaining exons were located in the IRs".

8) Line 242: Does "SC" written in this line refers to SSC region of chloroplast genome?

9) Line 315: Distribution of the SSRs loci in chloroplast genome can also be mentioned giving the details of their location in coding region or intergenic spacer region in tabular form would be helpful

10) Line 392: Please correct the spelling of freedom it's written "freedon".

11) Line 437: Please rewrite the description of Fig 7 in little more detail.

12) Line 590: Reference number 43 is not written in the prescribed format.

Reviewer #2: This study reports the complete plastome of Siraitia grosvenorii, a traditional chinese medicine with antidiabetic, antioxidative and anti inflammatory properties and Siraitia siamensis, its wild relative, having 158751 bp and 159190 bp respectively. High resolution novel molecular markers GSPC-F/R, GSPR-F/R, GSPB-F/R and GSPY-F/R were developed to distinguish between the two species. Phylogenetic analysis of the two species within the Cucurbitaceae family placed the two species close to M. kirilowii

Minor Comments

Material and Methods

Chloroplast genome assembly:

1) Mention the parameters used while assembling using SPAdes.

2) Why specific de-novo genome assembler like NOVOPlasty which is specific for plant chloroplast genome was not used.

3) How are the results different if specific genome assemblers are used

Genome annotation, repeats and simple sequence repeats (SSRs) analyses:

1) DOGMA is no longer supported. Annotate using newer plastid annotation tools.

Reviewer #3: First of all, l would like to acknowledge Dr Shashi Kumar, Aademic Editorail Board Membeanting member of PLOS One, for providing me this Scholastic Opportunity to peer review, and His patience as well as forbearance whilst receiving my Reviewer Comments. At the outset, I am glad to congratulate the Respected authors from Chinese Academy of Medical Sciences & Peking Union College Beijing, PRC: Drs. Xiaojun Ma, Hongwu Shi, Meng Yang, Changming Mo, Wenjuan Xie, Chang Liu and Bin Wu for having worked assiduously on such a significant paper on Chroloplast de No asssembly and sequence anaysis, with deeply underpinned significane in Medicinal Plant Breeding.

This research artTcle is worthy to be accepted for publication in PLOS ONE, since it satisfies each of the below Criteria:

1. The study presents the results of original research.

2. Results reported have not been published elsewhere.

3. Experiments, statistics, and other analyses are performed to a high technical standard and are described in sufficient detail.

4. Conclusions are presented in an appropriate fashion and are supported by the data.

5. The article is presented in an intelligible fashion and is written in standard English.

6. The research meets all applicable standards for the ethics of experimentation and research integrity.

7. The article adheres to appropriate reporting guidelines and community standards for data availability.

(i) The 333bp disparity between Chloroplast genome assemblies of grosvenorii and siamensis can be carefully addressed using Compositional data analysis paradigm for checking INTEGRITY of genome assemblies carried out thus far expected to be published by the Peer Reviewer soon in NAR-Genomics Bioinformtics CoDA issue CfP. The authors are kindly requested to revisit in this regard, my works on FastQ-ome (15th SocBiN, Moscow, Russia) and IUPAC-IUB probability conservatioBan (DFG- Hyderabad).

(ii) The 'Cucurbitaceae' phylogenetic divergence lineage is suggested to be revised and precisely expressed in terms of Mya or Bya using Carbon-dating.

(iii) The sweetness indices may be more favorably expressed in terms of Aspartame, an artifical sweetner, which is ~400 times sweeter than Sucrose.

(iv) Chinese Pharmacopoeia and Guangxi Medical Botanical Garden of IMPLAD can please share the 2 Germplasm accessions with NBPGR, National Bureau of Plant Genetic Resources, New Delhi (India) for better data interoperability. Also, apart from GenBank/ NCBI, kindly deposit and archive in Gigascience as well. Hopefully, RefSeq assemblies shall be annotated soon, with Feature table flat files made available, such as rRNA, tRNA, CDS.

(v) The two INDEL markers and 3 Aconitum species may be plotted in a Ternary diagram for ease of Visualisation for readers.

(vi) Apart from Trimmomatic for the Adapter removal, fastx-clipper and Cutadapt may be employed, for comparative validation, fastqc_data ext files may be autoparsed for text-mining the adapter sequences using Natural Language Processing from master Adapter files, such as Illumina Universal Sequence adapters.

(vii) SRA-BLASTN may be preferably mentioned over simply BLASTN under "Chloroplast genome assembly" section, since Illumina paired end reads are the BLAST database.

(viii) Apart from SPAdes and CLC Genomics Workbench, minia developed by Rayan Chikki, Inria (France) can also please be used since it is Computationally efficient.

(ix) SSR analysis may be re-performed in line with Peer Reviewer's FAO, Rome abstract communicated to 8th International Conference on Agricultural Statistics, IASRI New Delhi, toward which all the Chinese authors are cordially inviited to attend my presentation, as a self-cited extension of my Indo-Belarusian and 2nd IOCBS,Kolkata works on k-Mer based permutative Confusion matrix analysis of mono-, di-, tri-, tetra- nucleotide frequencies with Five number summary by Sourmash, jellyfish.a

(x) Relative Synonymoous Codon Usage (RCSU) may be repercieved in the Light of Chargaff's rules and Wobble hypothesis. Refer to my Big Data Genetic Code bit.ly/bdgcode

(xi) Coming to Hamming distane allowance of 30, apart from REPuter, RADMIDs subtool from RADtools (UK) may be implemented, even for deNovo SNP genotyping in the Chloroplast genomes.

(xii) MAFFT fourier transform-based Multiple Sequence Alignment of the 64 Protein-coding gene sequences may be complemened by Low Complexity Regions-MSA analysis akin to My DST-NNMCB proceeding derivative work of Rune Lindin Lab, Denmark Kinome Rendering Work furthered during my Pre Doctorate at SggW, Poland during Erasmus Mundu

(xiii) Intron-Exon distributions in clpP, ycf3 genes may be set theoretically analysed using Venny plots. Partial sequence overlaps in gene-Codon pairs, may be visualized using SPAdes ICARUS broswer from Center for Algorithmic Bioinformatics, SPbU Russia, since SPAdes was already run anyway.

(xiv) Contractionn-Expansion of IR regions may be modeled, off-bit using Compression-Rarefaction profiles inspired by Modern Acoustic Physics.

(xv) Strong AT/GC bias observations can neatly be further strengthened using Kurtosis calculations.

(xvi) Repeat structure can be subject to justification using pattern recognition as per Peer Reviewer's communication to Dr Liela Taher, University of Nurenbeg, Germany based upon Elementary Cellular Automata.

(xvii) Sequence divergence can be modeled using Entropy based decision trees as per my Seminar at Polish Academy of Sciences: Nencki Institute on SuperReference genome

(xviii) Specificity of Medicinal and Wild type molecular markers can be assessed better by Peer Reviewer's Octa filial homozygosity computations Schema to be presented Orally at IRRI, Phillipines Germplasm to Genome Conference at NASC, New Delhi : to which the All Authors and their research community are being cordially invited.

(xix) Selective pressure events may be better deciphered using Gene Expression studies, Both trancriptomic and small RNA interference, under Drought and Stress levels.

(xx) Trancription Factor DNA motif analysis can be confirmed using JASPAR database, ChIP seq approaches using MAC14 and gem-Tools at Bioinformatics level, and computationally augmented by Molecular Information Theory conception developed by Dr Thomas D. Schneider at NCI, National Cancer Institute, USA.

(xxI) Molecular markers at a distinguishable level for S.grosvenorii and S.siamenis can be ambitiously characterized by Tagging with Biodegradeable IoT/ Internet of Things, please refer to my TEDx talk delivered 2015 at Skopje, Former Yugoslavian Republic of Macedonia.

(xxii) Primer identification of the molecular markers can be inten-silico cross validated usiing Primer-BLAST and ApoE for instance.

(xxiii) The Four Molecular Markers GSPC-F/R, GSPR-F/R, GSPB-F/R, GSPY-F/R can be fitted into Confusion Matrix parameters namely True positives, Negatives and False Positives, Negatives so as to successfully subject the Binary classfication of the two Chloroplast species in a Mathematically coherent manner. Thank you.

6. PLOS authors have the option to publish the peer review history of their article (what does this mean?). If published, this will include your full peer review and any attached files.

Reviewer #1: Yes: Surbhi sharma

Reviewer #2: No

Reviewer #3: Yes: PRAHARSHIT SHARMA

---

## [Author Response · Author response to Decision Letter 0]

24 Oct 2019

Dear Dr Shashi Kumar:

We would like to thank you and three reviewers for the time in reviewing the manuscript "Complete chloroplast genomes of two Siraitia Merrill species: comparative analysis, positive selection and novel molecular marker development" (Manuscript Number: PONE-D-19-14994), and for providing us the opportunity to revise our manuscript. We have now improved the manuscript by adding new supporting information according to reviewer’s suggestions. The reviewer #3 is good at bioinformatics, who put forward many constructive comments, including many softwares and genetic algorithms. In our study, the softwares used were issued and ciated by the authoritative journal, and we did not aim to compare the difference among the software and algorithms. Anyway, we did our best to feed back the third reviewer’s suggestions. Below please find our point-to-point responses to the reviewers’ comments.

We are looking forward to hear from your regarding our revised manuscript.

Best regards, 

Xiaojun Ma, Professor

Institute of Medicinal Plant Development, Chinese Academy of Medical Science & Peking Union Medical College 

Phone: (+86) 010-5783-3155

Email: mayixuan10@163.com

Reviewer #1: The paper assembles and analyses two complete chloroplast genomes of the Siraitia genus. Paper covers most of the aspects of comparative genome analysis between the two Siraitia genus and have developed molecular markers to distinguish the two species. Overall the paper is well written, however there are few suggestions mentioned below:

1) Line 46: There is no reference to the statement written. Please add reference to the this statement.

Response: Thanks for the suggestion. We have added a reference to the statement.

2) Line 56: "Unstable treatment effect" statement is not clear.

Response: We thank the reviewer’s suggestion. “Unstable treatment effect" has been corrected as “poor therapeutic effect”

3) Line 62: "most origin plants" which origin plants author is mentioning, statement is not clear. 

Response: The sentence “To reduce the proportion of staminiferous plants and plant viral disease, most of the species have been collected and are cultivated via tissue culture to improve production, whereas most origin plants were obtained privately from wild resources without professional identification and named with ordinary variety names.” has been revised as “Most of the Siraitia plants origin privately from wild resources without professional identification and named with ordinary variety names.”

4) Line 168: Please mention the version of Modeltest used.

Response: We have added the version 3.7 to the Modeltest.

5) Line 197: Please add some description of the figure shown " The quadripartite structure includes two copies of an IR region (IRA and IRB) that separate large single-copy (LSC) and small single copy (SSC) regions ".

Response: According to reviewer's suggestion, we have described the figure “ Circular Gene map of the complete chloroplast genomes of S. grosvenorii and S. siamensis. The quadripartite structure includes two copies of an IR region (IRa and IRb) that separated by (LSC) and SSC regions. Genes drawn in the circle are the transcribed clockwise, and those on the outside are transcribed counter-clockwise. The darker gray area in the inner circle show the GC content, whereas the lighter corresponds to AT content. Different genes groups are colored.” .

6) Line 199: It would be easier if you could represent the chloroplast genome in linear form also for better and faster visualization.

Response: Thank for the suggestion. We have added a “S1 Figure” in the supporting information, which display the chloroplast genome in linear form. It is known that chloroplast genome is a single circular molecule with a typical quadripartite structure, therefore we retain the circular molecule structure in the main manuscript.

7) Line 230: Add reference of "S5 Table" next to line " In addition, the rps12 gene contained three exons and one intron because of trans-splicing, which resulted in a 5’ end exon located in the LSC region, whereas the remaining exons were located in the IRs".

Response: It is a good idea. The position of the "S5 Table" has been corrected.

8) Line 242: Does "SC" written in this line refers to SSC region of chloroplast genome?

Response：The “SC” means “single-copy”. We have decribed SC in the paragraph of Chloroplast Genome Assembly in the Materials and Methods as “the four boundaries between the single-copy (SC) regions and IR regions…”.

9) Line 315: Distribution of the SSRs loci in chloroplast genome can also be mentioned giving the details of their location in coding region or intergenic spacer region in tabular form would be helpful.

Response: Thanks for the suggestion. We have added a S9 Table (XLSX, Distribution of the SSRs loci in chloroplast genome of S. grosvenorii and S. siamensis.) in the supporting information which contain SSRs information of the two species in detail.

10) Line 392: Please correct the spelling of freedom it's written "freedon".

Response: The mistake has been corrected.

11) Line 437: Please rewrite the description of Fig 7 in little more detail.

Response: The description of Fig 7 has been revised as “Fig 7. Schematic diagram displayed the four novel molecular markers. The indel makers in the intergenic spacers, including GSPC-F/R, GSPR-F/R, and SCPB-F/R, and ycf1 SNP were verified in Siraitia species with five individuals. (A) Sequencing results showed that PCR amplification by GSPC-F/R between the ndhC and trnV-UAC in S. grosvenorii and S. siamensis were 790 bp and 808 bp, respectively. The difference can also be found by GSPR-F/R (B), and SCPB-F/R (C) between the two Siraitia species. (D) GSPY-F/R marker for ycf1 SNP sites were validated in each cases, which were effective to discriminate the two Siraitia species.”

12) Line 590: Reference number 43 is not written in the prescribed format.

Response: The reference has been corrected as “Beier S, Thiel T, Münch T, Scholz U, Mascher M (2017) MISA-web: a web server for microsatellite prediction. Bioinformatics 33: 2583-2585. doi: 10.1093/bioinformatics/btx198.”

Reviewer #2: This study reports the complete plastome of Siraitia grosvenorii, a traditional chinese medicine with antidiabetic, antioxidative and anti-inflammatory properties and Siraitia siamensis, its wild relative, having 158751 bp and 159190 bp respectively. High resolution novel molecular markers GSPC-F/R, GSPR-F/R, GSPB-F/R and GSPY-F/R were developed to distinguish between the two species. Phylogenetic analysis of the two species within the Cucurbitaceae family placed the two species close to M. kirilowii.

Minor Comments

Material and Methods

Chloroplast genome assembly:

1) Mention the parameters used while assembling using SPAdes.

Response: Thanks for the suggestions. The parameters has been added in the manuscript. The SPAdes using for the assembling that the parameters were set as “-k 21,33,55,77,99,127 –careful”. 

2) Why specific de-novo genome assembler like NOVOPlasty which is specific for plant chloroplast genome was not used.

Response: Thanks for the suggestion. In this study, the raw data were acquired and analyzed in march, 2017, while NOVOPlasty was not widely applied in the chloroplast genomen assembly at that time. After reading the comments, we download the paper by Nicolas et al. (NOVOPlasty: De novo assembly of organelle genomes from whole genome data), and knew that NOVOPlasty is an effective genome assembler for plant chloroplast genome. Therefore, we used NOVOPlasty to assemble the two species chloroplast genomes for validation. Result showed that the assembly by NOVOPlasty was identical to that of by SPAdes and CLC Genomics Workbench, indicating the reliability of our assembly.

3) How are the results different if specific genome assemblers are used.

Response: It only took about several hours for us to finish the assembly of the two Siraitia chloroplast genomes by NOVOPlasty. Result showed that the assembly by NOVOPlasty was identical to that of by SPAdes and CLC Genomics Workbench previously. Therefore, we will use NOVOPlasty for the chloroplast genome assembly of other plant species in the next work.

Genome annotation, repeats and simple sequence repeats (SSRs) analyses:

1) DOGMA is no longer supported. Annotate using newer plastid annotation tools.

Response: In this work, the chloroplast genome was premarily annotated by CPGAVAS, which is an effective software for the complete annotation of chloroplast genome. But a few annotions were blurry for CPGAVAS, which were re-annotated by DOGMA. Combined CPGAVAS with DOGMA, the chloroplast genomes of the two S. grosvenorii were annotated completely, and all the genes were blasted in the NCBI database for verification.

Reviewer #3: First of all, l would like to acknowledge Dr Shashi Kumar, Aademic Editorail Board Membeanting member of PLOS One, for providing me this Scholastic Opportunity to peer review, and His patience as well as forbearance whilst receiving my Reviewer Comments. At the outset, I am glad to congratulate the Respected authors from Chinese Academy of Medical Sciences & Peking Union College Beijing, PRC: Drs. Xiaojun Ma, Hongwu Shi, Meng Yang, Changming Mo, Wenjuan Xie, Chang Liu and Bin Wu for having worked assiduously on such a significant paper on Chroloplast de No asssembly and sequence anaysis, with deeply underpinned significane in Medicinal Plant Breeding.

This research article is worthy to be accepted for publication in PLOS ONE, since it satisfies each of the below Criteria:

1. The study presents the results of original research.

2. Results reported have not been published elsewhere.

3. Experiments, statistics, and other analyses are performed to a high technical standard and are described in sufficient detail.

4. Conclusions are presented in an appropriate fashion and are supported by the data.

5. The article is presented in an intelligible fashion and is written in standard English.

6. The research meets all applicable standards for the ethics of experimentation and research integrity.

7. The article adheres to appropriate reporting guidelines and community standards for data availability.

(i) The 333bp disparity between Chloroplast genome assemblies of grosvenorii and siamensis can be carefully addressed using Compositional data analysis paradigm for checking INTEGRITY of genome assemblies carried out thus far expected to be published by the Peer Reviewer soon in NAR-Genomics Bioinformtics CoDA issue CfP. The authors are kindly requested to revisit in this regard, my works on FastQ-ome (15th SocBiN, Moscow, Russia) and IUPAC-IUB probability conservatioBan (DFG- Hyderabad).

Response: Thanks for the suggestions. The assembly of the two chloroplast genomes were verified by specific de-novo genome assembler NOVOPlasty, and the boundary sequences were validated by PCR and suquencing, which indicated the integrity of the genome. We are glad to revisit the reviewer’s work on FastQ-ome and get lot from the paper “A random forest ensemble of FastQ reads as decision trees”.

(ii) The 'Cucurbitaceae' phylogenetic divergence lineage is suggested to be revised and precisely expressed in terms of Mya or Bya using Carbon-dating.

Response: Thanks for the suggestion. Phylogenetic divergence lineage could be expressed in different methods. Xia et al reported the S. grosvenorii divergened from menbers of the Cucurbitaceae family at approximately 40.9 million years ago based on the phylogenetic analysis (DOI: 10.1093/gigascience/giy067). In this manuscript, we constructed the 'Cucurbitaceae' phylogenetic divergence lineage in the molecular level based on coding-sequence in the chloroplast genomes, which contained conservative and divergent sequences. It is not the focus for us to constructe the phylogenetic divergence lineage in terms of Mya or Bya.

(iii) The sweetness indices may be more favorably expressed in terms of Aspartame, an artifical sweetner, which is ~400 times sweeter than Sucrose.

Response: We agree the standpoint. Aspartame is more favorably for display the sweetness indices. For the authenticity of the quotation, we corrected it as “and is approximately 560 times sweeter than sucrose, and is about 1.4 fold sweeter than aspartame”.

(iv) Chinese Pharmacopoeia and Guangxi Medical Botanical Garden of IMPLAD can please share the 2 Germplasm accessions with NBPGR, National Bureau of Plant Genetic Resources, New Delhi (India) for better data interoperability. Also, apart from GenBank/ NCBI, kindly deposit and archive in Gigascience as well. Hopefully, RefSeq assemblies shall be annotated soon, with Feature table flat files made available, such as rRNA, tRNA, CDS.

Response: Thanks for the suggestion. Firstly, we are pleased to share the 2 germplasm accessions with NBPGR (India) for better data interoperability within the scope of laws and regulations in P.R.C.. Secondly, we have uploaded the data of the two chloroplast genomes sequences on the GenBank/ NCBI with annotation, including the rRNA, tRNA and CDS. When the manuscript is be accepted, we will release all the data. We will deposit the data in Gigascience until the manuscript are published.

(v) The two INDEL markers and 3 Aconitum species may be plotted in a Ternary diagram for ease of Visualisation for readers.

Response: It is a good idea to plot the two INDEL markers and 3 Aconitum specie in a Ternary diagram for the ease of visualisation for readers. We would use this display method in the latter writing. In this manuscript, the “Two INDEL markers and 3 Aconitum species” in the Introduction was summarized others results. If the reader want to learn more details, they could read the original article.

(vi) Apart from Trimmomatic for the Adapter removal, fastx-clipper and Cutadapt may be employed, for comparative validation, fastqc_data ext files may be autoparsed for text-mining the adapter sequences using Natural Language Processing from master Adapter files, such as Illumina Universal Sequence adapters.

Response: Thanks for the reviewer’s suggestion. It was known that Trimmomatic was widely used for the Adapter removal, of raw seqnencing data and related article have been published in authoritative journals, such as Nature Protocols (DOI:10.1038/nprot.2016.011 ) and Genome Biology (DOI: 10.1186/s13059-014-0517-9). Both fastx-clipper and Cutadapt are all-right softwares for adapter removal, and Illumina Universal Sequence adapters are good for solving the comparative validation. We will improve our methods in the latter research. 

(vii) SRA-BLASTN may be preferably mentioned over simply BLASTN under "Chloroplast genome assembly" section, since Illumina paired end reads are the BLAST database.

Response: Thanks for the suggestion. We are sorry that the methods of "Chloroplast genome assembly" writen in the manuscript was simply, which had benn improved. We have corrected the “BLASTN” with “SRA-BLASTN”. 

(viii) Apart from SPAdes and CLC Genomics Workbench, minia developed by Rayan Chikki, Inria (France) can also please be used since it is Computationally efficient.

Response: Thanks for the suggestion. Apart from SPAdes and CLC Genomics Workbench, we also used specific de-novo genome assembler NOVOPlasty, which is specific for plant chloroplast genome. We will adopt the Minia for the further research, because it is also a good software.

(ix) SSR analysis may be re-performed in line with Peer Reviewer's FAO, Rome abstract communicated to 8th International Conference on Agricultural Statistics, IASRI New Delhi, toward which all the Chinese authors are cordially invited to attend my presentation, as a self-cited extension of my Indo-Belarusian and 2nd IOCBS, Kolkata works on k-Mer based permutative Confusion matrix analysis of mono-, di-, tri-, tetra- nucleotide frequencies with Five number summary by Sourmash, jellyfish.a

Response: Thanks for the suggestions. We have re-performed the SSR analysis and added S9 Table (XLSX) in supporting information, which described distribution of the SSRs loci in chloroplast genomes in details.

(x) Relative Synonymoous Codon Usage (RCSU) may be repercieved in the Light of Chargaff's rules and Wobble hypothesis. Refer to my Big Data Genetic Code bit.ly/bdgcode

Response: Thanks for the suggestion. We analyzed the Relative Synonymoous Codon Usage (RCSU) by the software CodonW (1.4.4). The programme CodonW is written in standard ANSI C, and it compiles cleanly using the GNU C compiler (version 2.7.2.1) with the stringent ANSI and pedantic command line switches. In addition, Cusp and Compseq in EMBOSS (v.6.3.1) were used for the analysis of codon usage frequency and GC content. Bioinformation analysis by the programme CodonW, Cusp and Compseq in EMBOSS has been used in, many paper, such as ‘Weinel C et al. General method of rapid Smith/Birnstiel mapping adds for gap closure in shotgun microbial genome sequencing projects: application to Pseudomonas putida KT2440. Nucleic Acids Research, 2001, 29(22):E110.

(xi) Coming to Hamming distane allowance of 30, apart from REPuter, RADMIDs subtool from RADtools (UK) may be implemented, even for deNovo SNP genotyping in the Chloroplast genomes.

Response: Thanks for the suggestions. In this manuscript, REPuter software was used to analysis the Repeats, including forward, palindromic, reverse, and complement. REPuter is a authoritative software for repeats analysis as decribed in the paper (‘REPuter: fast computation of maximal repeats in complete genomes’, DOI: 10.1093/bioinformatics/15.5.426). In the recent years, REPuter was widely used in some medicinal plant chloroplast genome analysis, and was cited frequently. We also believe that the RADMIDs subtool from RADtools is an effective for repeats analysis. It is known that very software own superiorities and limitations. The RADMIDs subtool would be used in our further study. 

(xii) MAFFT fourier transform-based Multiple Sequence Alignment of the 64 Protein-coding gene sequences may be complemented by Low Complexity Regions-MSA analysis akin to My DST-NNMCB proceeding derivative work of Rune Lindin Lab, Denmark Kinome Rendering Work furthered during my Pre Doctorate at SggW, Poland during Erasmus Mundu

Response: Thanks for the suggestion. In our manuscript, The MAFFT fourier transform-based Multiple Sequence Alignment of 64 consensus protein-coding gene sequences was complemented by Low Complexity Regions-MSA analysis . MAFFT alignment was used in many species for the phylogenetic analyses of chloroplast genomes, and was issued in Nucleic Acids Research titled with ‘aLeaves facilitates on-demand exploration of metazoan gene family trees on MAFFT sequence alignment server with enhanced interactivity’ (DOI: 10.1093/nar/gkt389). 

(xiii) Intron-Exon distributions in clpP, ycf3 genes may be set theoretically analysed using Venny plots. Partial sequence overlaps in gene-Codon pairs, may be visualized using SPAdes ICARUS broswer from Center for Algorithmic Bioinformatics, SPbU Russia, since SPAdes was already run anyway.

Response: Thanks for the suggestion. Intron-Exon distributions in clpP, ycf3 genes and some other gene contained introns and exons could be found in Fig 4. Partial sequence overlaps in gene-Codon pairs could also be visualized in Fig 4.

(xiv) Contraction-Expansion of IR regions may be modeled, off-bit using Compression-Rarefaction profiles inspired by Modern Acoustic Physics.

Response: This is a good idea. The Compression-Rarefaction profiles inspired by Modern Acoustic Physics is creative. We want to have a try to using this method for Contraction-Expansion of IR regions model, although our team don’t have this experience. If possible, we are pleasured to learn and discuss it with you.

(xv) Strong AT/GC bias observations can neatly be further strengthened using Kurtosis calculations.

Response: Thank you for the recommendation. The Kurtosis calculations is a software for deal with mass data with gaussian distribution. In our study, GC content in the four regions (LSC, SSC, IRa, and IRb) were analyzed, and two species had approximately 36.8% GC. We attempt to analysis the AT/GC bias by Kurtosis calculations, whereas it was difficult to assigned the parameters for the four bases, and there is litter reference for the AT/GC bias using Kurtosis calculations. 

(xvi) Repeat structure can be subject to justification using pattern recognition as per Peer Reviewer's communication to Dr Liela Taher, University of Nurenbeg, Germany based upon Elementary Cellular Automata.

Response: Thanks for the suggestion. In our manuscript, repeat structure was identified by REPuter software. We think that all the result from the bioinformatical analysis should be justified with experiments before applied. The pattern recognition might make the analysis results get more accurate than analysis by REPuter software. If necessary, we will apply the pattern recognition for the Repeat structure until it is meaningful in application.

(xvii) Sequence divergence can be modeled using Entropy based decision trees as per my Seminar at Polish Academy of Sciences: Nencki Institute on SuperReference genome

Response: Entropy based decision trees might be a new approach for the Sequence divergence analysis. We think it would be visual with the modeled analysis. We want to learn how to make the Sequence divergence modeled because of the lack of details for your Seminar at Polish Academy of Sciences.

(xviii) Specificity of Medicinal and Wild type molecular markers can be assessed better by Peer Reviewer's Octa filial homozygosity computations Schema to be presented Orally at IRRI, Phillipines Germplasm to Genome Conference at NASC, New Delhi : to which the All Authors and their research community are being cordially invited.

Response: Thanks for the suggestion. The homozygosity computations Schema for the assessments of specificity of medicinal and wild type molecular markers is a creative method. We will pay attention to the conference at NASC, New Delhi, and the molecular markers developed form the two Siraitia species chloroplast genomes will be assessed with the homozygosity by computational approach in the future.

(xix) Selective pressure events may be better deciphered using Gene Expression studies, Both transcriptomic and small RNA interference, under Drought and Stress levels.

Response: We agree with the points. The transcriptomic and small RNA interference are important approach for deciphering the selective pressure events. In the further work, we will use these methods to verify the selective pressure events under drought and stress levels. In this manuscript, all the analysis were based on the genome level.

(xx) Transcription Factor DNA motif analysis can be confirmed using JASPAR database, ChIP seq approaches using MAC14 and gem-Tools at Bioinformatics level, and computationally augmented by Molecular Information Theory conception developed by Dr Thomas D. Schneider at NCI, National Cancer Institute, USA.

Response: Thanks for the review’s suggestions. Transcription Factor DNA motif analysis is not the focus in this work. We will improve our work with these methods in the future.

(xxi) Molecular markers at a distinguishable level for S. grosvenorii and S. siamenis can be ambitiously characterized by Tagging with Biodegradeable IoT/ Internet of Things, please refer to my TEDx talk delivered 2015 at Skopje, Former Yugoslavian Republic of Macedonia.

Response: Thank you for the suggestion. In this study, we developed four molecular markers to distinguish the S. grosvenorii and S. siamensis. We are desired to explore more Siraitia resources to exam the availability of the four molecular markers. Finally, we will make the markers by Tagging with Biodegradeable IoT/ Internet of Things.

(xxii) Primer identification of the molecular markers can be inten-silico cross validated usiing Primer-BLAST and ApoE for instance.

Response: We agreed with the viewpoint. In order to insure the accuracy of markers, these markers were validated by PCR and sequenced.

(xxiii) The Four Molecular Markers GSPC-F/R, GSPR-F/R, GSPB-F/R, GSPY-F/R can be fitted into Confusion Matrix parameters namely True positives, Negatives and False Positives, Negatives so as to successfully subject the Binary classfication of the two Chloroplast species in a Mathematically coherent manner. Thank you.

Response: In our study, all the four molecular markers GSPC-F/R, GSPR-F/R, GSPB-F/R, GSPY-F/R were validated the truth by the two species with different individuals. As shown in Fig 7, the indel makers in the intergenic spacers, including GSPC-F/R, GSPR-F/R, and SCPB-F/R, and ycf1 SNP were verified in Siraitia species with five individuals. Sequencing results showed that PCR amplification by GSPC-F/R between the ndhC and trnV-UAC in S. grosvenorii and S. siamensis were 790 bp and 808 bp, respectively. The difference can also be found by GSPR-F/R, and SCPB-F/R between the two Siraitia species. GSPY-F/R marker for ycf1 SNP sites were validated in each case, which were effective to discriminate the two Siraitia species.

---

## [Decision Letter · Decision Letter 1]

10 Dec 2019

Complete chloroplast genomes of two Siraitia Merrill species: comparative analysis, positive selection and novel molecular marker development

PONE-D-19-14994R1

Dear Dr. Xiaojun Ma,

We are pleased to inform you that your manuscript has been judged scientifically suitable for publication and will be formally accepted for publication once it complies with all outstanding technical requirements.

With kind regards,

Shashi Kumar, Ph.D.

Academic Editor

PLOS ONE

Reviewers' comments:

Reviewer's Responses to Questions

**Comments to the Author**

1. If the authors have adequately addressed your comments raised in a previous round of review and you feel that this manuscript is now acceptable for publication, you may indicate that here to bypass the “Comments to the Author” section, enter your conflict of interest statement in the “Confidential to Editor” section, and submit your "Accept" recommendation.

Reviewer #1: All comments have been addressed

Reviewer #2: All comments have been addressed

2. Is the manuscript technically sound, and do the data support the conclusions?

Reviewer #1: Yes

Reviewer #2: Yes

3. Has the statistical analysis been performed appropriately and rigorously? 

Reviewer #1: Yes

Reviewer #2: Yes

4. Have the authors made all data underlying the findings in their manuscript fully available?

Reviewer #1: Yes

Reviewer #2: Yes

5. Is the manuscript presented in an intelligible fashion and written in standard English?

Reviewer #1: Yes

Reviewer #2: Yes

6. Review Comments to the Author

Reviewer #1: Author have incorporate the changes suggested in previous review which gives more clarity to the data and thus to the final study.

Reviewer #2: The authors have addressed all the comments raised in the previous version of the manuscript which has improved the manuscript considerably. The manuscript is technically sound and the data supports the conclusions drawn. All the statistical analyses have been rigorously performed. The manuscript meets PLOS ONE criteria and I am happy to accept this publication. Congratulations to the authors

7. PLOS authors have the option to publish the peer review history of their article (what does this mean?). If published, this will include your full peer review and any attached files.

Reviewer #1: No

Reviewer #2: No

---

## [Editor Report · Acceptance letter]

13 Dec 2019

PONE-D-19-14994R1 

Complete chloroplast genomes of two *Siraitia* Merrill species: comparative analysis, positive selection and novel molecular marker development 

Dear Dr. Ma:

I am pleased to inform you that your manuscript has been deemed suitable for publication in PLOS ONE. Congratulations! Your manuscript is now with our production department. 

With kind regards,

on behalf of

Dr. Shashi Kumar 

Academic Editor

PLOS ONE